# ATTACK GRAPH CONVOLUTIONAL NETWORKS BY ADDING FAKE NODES

## ABSTRACT

Graph convolutional networks (GCNs) have been widely used for classifying graph nodes in the semi-supervised setting. Previous work have shown that GCNs are vulnerable to the perturbation on adjacency and feature matrices of existing nodes. However, it is unrealistic to change existing nodes in many applications, such as existing users in social networks. In this paper, we design algorithms to attack GCNs by adding fake nodes. A greedy algorithm is proposed to generate adjacency and feature matrices of fake nodes, aiming to minimize the classification accuracy on the existing nodes. In additional, we introduce a discriminator to classify fake nodes from real nodes, and propose a Greedy-GAN attack to simultaneously update the discriminator and the attacker, to make fake nodes indistinguishable to the real ones. Our non-targeted attack decreases the accuracy of GCN down to 0.10, and our targeted attack reaches a success rate of 99% on the whole datasets, and 94% on average for attacking a single target node.

## 1 INTRODUCTION

Graphs play a very important role in many real world applications, such as social networks (Facebook and Twitter), biological networks (protein-protein interaction networks and gene interaction networks), as well as attribute graphs (PubMed and Arxiv) (Grover & Leskovec, 2016; Ying et al., 2018; Rhee et al., 2017). Node classification is one of the most important tasks on graphs—given a graph with labels associated with a subset of nodes, predict the labels for rest of the nodes. For this node classification task, deep learning models on graphs, such as Graph Convoltional Networks (GCNs), have achieved state of the art performance (Kipf & Welling, 2016). Moreover, GCNs have wide applications in cyber security, where they can learn a close-to-correct node labeling semi-autonomously. This reduces the load on security experts and helps to manage networks that add or remove nodes dynamically, such as, WiFi networks in universities and web services in companies.

The wide applicability of GCNs motivates recent studies about their robustness. Zügner et al. (2018) Dai et al. (2018) developed algorithms to attack GCNs, showing that by altering a small amount of edges and features, the classification accuracy of GCNs can be reduced to chance-level. However, changing edges or features associated with existing nodes is impractical in many cases. For example, in social network applications, an attacker has to login to the users' accounts to change existing connections and features, and gaining login accesses is almost impossible. In comparison, adding fake nodes that correspond to fake accounts or users, can be much easier in practice. But the key question is *can we interfere the classification results of existing nodes by adding fake nodes to the network?* We answer this question affirmative by introducing novel algorithms to design fake nodes that successfully reduce GCN's performance on existing nodes.

To design the adjacency and feature matrices associated with fake nodes, we have to address two challenges. First, the edges and features are usually discrete 0/1 variables. Although there have been many algorithms proposed for attacking image classifiers, such as FGSM, C&W and PGD attacks (Goodfellow et al., 2014; Carlini & Wagner, 2017; Madry et al., 2017), they all assume continuous input space and cannot be directly applied to problems with discrete input space. Second, it's not easy to make the fake nodes "looked" like the real ones? For example, if we add a fake node that connects to all existing nodes, the system can easily detect and disable such fake node. In this paper, we propose two algorithms, Greedy attack and Greedy-GAN attack, to address these two challenges. Our contributions can be summarized below:

- To the best of our knowledge, this is the first paper studying how to add fake nodes to attack GCNs. We do not need to manipulate existing nodes' adjacency and feature matrices.

- We propose a Greedy attack algorithm to address the discrete input space problem in designing fake nodes' adjacency and feature matrices.

- We introduce a discriminator to classify fake nodes from real nodes, and propose a Greedy-GAN algorithm to simultaneous optimize the discriminator and the attacker. Despite a lower successful rate, this approach can make fake nodes harder to detect.

- We conduct experiments on several real datasets. For non-targeted attack, we get accuracy down to 0.10 for the Cora dataset, and 0.14 for the Citeseer dataset. For targeted attack on whole datasets, Greedy attack have up to 99% success rate on Cora and 90% on Citeseer. For targeted attack on a single node, it could reach 94% success rate on Cora and 0.80% success rate on Citeseer.

## 2 RELATED WORK

**Adversarial Attacks** Adversarial examples for computer vision tasks have been studied extensively. Goodfellow et al. (2014) discovered that deep neural networks are vulnerable to adversarial attacks—a carefully designed small perturbation can easily fool a neural network. Several algorithms have been proposed to generate adversarial examples for image classification tasks, including FGSM (Goodfellow et al., 2014), IFGSM (Kurakin et al., 2016), C&W attack (Carlini & Wagner, 2017) and PGD attack (Madry et al., 2017). In the black-box setting, it has also been reported that an attack algorithm can have high success rate using finite difference techniques (Chen et al., 2017b), and several algorithms are recently proposed to reduce query numbers (Ilyas et al., 2018; Suya et al., 2017). Transfer attack Papernot et al. (2016a) and ensemble attack Tramèr et al. (2017) can also be applied to black-box setting, with lower successful rate but less number of queries. Besides attacking image classification, CNN related attacks have also been explored. A typical usage is attacking semantic segmentation and object detection (Metzen et al., 2017; Arnab et al., 2017; Xie et al., 2017; Lu et al., 2017; Eykholt et al., 2017). Image captioning (Chen et al., 2017a) and visual QA (Xu et al., 2017) could also be attacked.

Most of the above-mentioned work are focusing on problems with continuous input space (such as images). When the input space is discrete, attacks become discrete optimization problems and are much harder to solve. This happens in most natural language processing (NLP) applications. For text classification problem, Fast Gradient Sign Method (FGSM) is firstly applied by (Papernot et al., 2016b). Deleting important words (Li et al., 2016), replacing or inserting words with typos and synonyms (Samanta & Mehta, 2017; Liang et al., 2017). For black box setting, Gao et al. (2018) develops score functions to find out the works to modify. Jia & Liang (2017) adds misleading sentences to fool the reading comprehension system. Zhao et al. (2017) use GAN to generate natural adversarial examples. Ebrahimi et al. (2017) and Cheng et al. (2018) attacks machine translation system Seq2Seq by changing words in text.

**Graph Convolutional Neural Networks (GCN).** Classification of graph nodes has become an important problem in cyber security applications and recommender systems. Graph convolutional networks (GCNs) solve the problem in an end-to-end manner and can avoid the "optimize twice" problem of previous graph embedding methods. The main idea of GCNs is to aggregate associated information from a node and its neighbors using some aggregation functions. They train the aggregation steps and the final prediction layer end-to-end to achieve better performance than traditional approaches. There are several variations of GCNs proposed recently (Kipf & Welling, 2016; Pham et al., 2016; Defferrard et al., 2016; Hamilton et al., 2017; Chen et al., 2018; Ying et al., 2018), and we will focus on the commonly used structure proposed in (Kipf & Welling, 2016).

There have been several work on attacking GCNs. Recently, Zügner et al. (2018) published an attack on GCNs by changing current nodes' links and features. They present a FGSM-like method and optimize a surrogate model named Nettack to choose the edges and features that should be manipulated. Dai et al. (2018) showed that by only manipulating graph structure, the Graph Neural Networks are quite vulnerable to the attacks of challenging few edges. They proposed reinforcement learning-based attack method to attack. They also employed a gradient ascent method to change the

graph structures in the white-box setting. Both of these two papers perform non-targeted attacks only.

Instead of altering edges or features of existing nodes, we develop novel algorithms to add fake nodes to interfere the performance of GCNs. This has not been done in previous work. Furthermore, we test our algorithm in both targeted and non-targeted attacks in the experiments.

# 3 PRELIMINARY

GCN is a semi-supervised learning method to classify nodes in attribute graphs. Given an adjacency matrix $A \in \mathbb{R}^{n \times n}$, feature matrix $X \in \mathbb{R}^{n \times d}$, and a subset of labeled nodes, the goal is to predict the labels of all the nodes in the graph. There are several variations of GCNs, but we consider one of the most common approaches introduced in (Kipf & Welling, 2016). Starting from $H^0 = X$, GCN uses the following rule to iteratively aggregate features from neighborhoods:

$$H^{(l+1)} = \sigma(\tilde{D}^{-\frac{1}{2}} \tilde{A} \tilde{D}^{-\frac{1}{2}} H^{(l)} W^{(l)}) \tag{1}$$

where $\tilde{A} = A + I_N$ is the adjacency matrix of the undirected graph with added self connections, $I_N$ is the identity matrix, $\tilde{D}$ is a diagonal matrix with $\tilde{D}_{i,i} = \sum_j \tilde{A}_{ij}$, and $\sigma$ is the activation function. We set $\sigma(x) = \text{ReLU}(x) = \max(0, x)$, which is the most common choice in practice. For a GCN with $L$ layers, after getting the top-layer feature $H^L$, a fully connected layer with soft-max loss is used for classification. A commonly used application is to apply two-layer GCN for semi-supervised learning node classification on graph (Kipf & Welling, 2016). The model could be simplified as:

$$Z = f(X, A) = \text{softmax}(\hat{A} \sigma(\hat{A} X W^{(0)}) W^{(1)}), \tag{2}$$

where $\hat{A} = \tilde{D}^{-\frac{1}{2}} \tilde{A} \tilde{D}^{-\frac{1}{2}}$. Another choice for forming $\hat{A}$ is to normalize the adjacency matrix by rows, leading to $\hat{A} = \tilde{D^{-1}} \tilde{A}$. We will experience with both choices in the experimental results.

For simplicity, we will assume our target network is structured as equation 2, but in general our algorithm can be used to attack GCNs with more layers.

# 4 ATTACK ALGORITHMS

In this section, we will describe our "fake nodes" attack on GCNs. We will describe both a non-targeted attack, which tries to lower the accuracy of all the existing nodes uniformly, and a targeted attack, which attempts to force the GCN to give a desired label to nodes. Instead of manipulating the feature and adjacency matrices of existing nodes, we insert $m$ fake nodes with corresponding fake features into the graph. After that, the adjacency matrix is $A' = \begin{bmatrix} A & B^T \\ B & C \end{bmatrix}$ and the feature matrix becomes : $X' = \begin{bmatrix} X \\ X_{fake} \end{bmatrix}$. Note that $A$ is the original adjacency matrix and $X$ is the original feature matrix. Starting from $B = 0, C = I$, our goal is to design $B, C, X_{fake}$ to achieve the desired objective (e.g., lower the classification accuracy on existing nodes).

## 4.1 NON-TARGETED ATTACK

The goal of non-targeted attack is to lower the classification accuracy on all the existing nodes by designing features and links of fake nodes. We use the accuracy of GCN to measure the effectiveness of attacks. We will present two different algorithms to attack GCNs: Greedy attack that updates links and features one by one, and Greedy-GAN attack that uses a discriminator to generate unnoticeable features of fake nodes.

### 4.1.1 GREEDY ATTACK

In our attack, we define the objective function as

$$J(A', X') = \sum_{i=1,...,n} \left( \max \left( [\hat{A}' \sigma(\hat{A}' X W^{(0)}) W^{(1)}]_{i,:} \right) - [\hat{A}' \sigma(\hat{A}' X W^{(0)}) W^{(1)}]_{i,y_i} \right), \tag{3}$$

---

**Algorithm 1** Greedy Attack

---

**Input:** Adjacency matrix $A$; feature matrix $X$; A classifier $f$ with loss function $J$; number of iterations $T$.
**Output:** Modified graph and features $G' = (A', X')$ after adding fake nodes.
**for:** t = 0 to $T - 1$ **do**
    **Let** $e^* = (u^*, v^*) \leftarrow \arg\max \nabla_{B,C} J(A', X')$
        $G_{B,C}^{(t+1)} \leftarrow G_{B,C}^{(t)} + e^*$
    **Let** $f^* = (u^*, i^*) \leftarrow \arg\max \nabla_{X_{fake}} J(A', X')$
        $G_{X_{fake}}^{(t+1)} \leftarrow G_{X_{fake}}^{(t)} + f^*$
**return:** $G^{(t)}$

---

where $y_i$ is the correct label of node $i$. In this objective function, if the largest logit of node $i$ is not the correct label $y_i$, it will encounter a positive score; otherwise the score will be zero. We then solve the following optimization problem to form the fake nodes:

$$\arg\max_{B,C,X_{fake}} J(A', X') \quad \text{s.t.} \quad \|B\|_0 + \|C\|_0 + \|X_{fake}\|_0 \leq T, \tag{4}$$

where $\| \cdot \|_0$ denotes number of nonzero elements in the matrix. Also, we assume $B, C, X_{fake}$ can only be 0/1 matrices. Unlike images, graphs have discrete values in the adjacency matrix, and in many applications the feature matrix comes from indicator of different categories. Therefore, gradient-based techniques such as FGSM and PGD cannot be directly applied.

Instead, we propose a greedy approach—starting from $B, C, X_{fake}$ all being zeros, we add one feature and one edge at each step. To add a feature, we find the the maximum element in $\nabla_{X_{fake}} J(A', X')$ and turn it into nonzero. Similarly, we find the maximum element in $\nabla_{B,C} J(A', X')$ and add the entry to the adjacency matrix. The Greedy attack is presented in Algorithm 1.

In the algorithm, when adding links and features, we make sure that there is no such a link or feature before adding. In practice, we can adjust the frequency of feature and weight updates. For example, if the original adjacency matrix has twice nonzero elements than the feature matrix, we can update two elements in the adjacency matrix and one element in the feature matrix at every two iterations.

### 4.1.2 GREEDY-GAN ATTACK

Next we will present the attack based on the idea of Generative Adversarial Network (GAN). The main idea is to add a discriminator to generate fake features that are similar to the original ones. In order to do this, we first design a neural network with two fully connected layers plus a softmax layer as the discriminator, which can be written as

$$D(X') = \text{softmax}(\sigma(X'W^{(0)})W^{(1)}), \tag{5}$$

where softmax works on each row of the output matrix. Each element in $D(X')$ indicates whether the discriminator classifies the node as real or fake.

We want to generate fake nodes with features similar to the real ones to fool the discriminator. Since the output of discriminator is binary, we use binary cross entropy loss defined by $L(p, y) = -(y \log(p) + (1-y) \log(1-p))$, where $y$ is binary indicator of the ground-truth (real or fake images), and $p$ is the predicted probability by our discriminator. Then we solve the following optimization problem :

$$\arg\max_{B,C,X_{fake}} \min_D (J(A', X') - c * L(D(X'), Y)) \tag{6}$$

where $Y$ is the ground-truth (real/fake) indicator for nodes and $c$ is the parameter determine with the weight of discriminator and the GCN performance. For example, if $c$ is set with a very large value, the objective function is dominated by the discriminator, so the node features generated will be very close to real but with lower attack successful rate.

We adopt the GAN-like framework to train both features/adjacency matrices and discriminator parameters iteratively. In experiments, at each epoch we conduct $10,000$ greedy updates for

---

**Algorithm 2** Greedy-GAN Attack

---

**Input:** Adjacency matrix $A$; feature matrix $X$; A classifier $f$ with loss function $J$; Discriminator $D$ with loss function $L$; number of iterations $T$.

**Output:** Modified graph $G' = (A', X')$ after adding fake nodes.

**for:** $t = 0$ to $T - 1$ **do**

    **if** $t \% 10000 == 0$:

        **retrain** discriminator $D$ 10 times.

    Let $e^*_{add} = (u^*_{add}, v^*_{add}) \leftarrow \arg\max \nabla_{B,C}[J(A', X') - c * L(D(X'))]$

    $e^*_{drop} = (u^*_{drop}, v^*_{drop}) \leftarrow \arg\min \nabla_{B,C}[J(A', X') - c * L(D(X'))]$

    **if** $|\nabla_{B,C}[J(A', X') - c * L(D(X'))]_{e^*_{add}}| > |\nabla_{B,C}[J(A', X') - c * L(D(X'))]_{e^*_{drop}}|$ **:**

        $G^{(t+1)}_{B,C} \leftarrow G^{(t)}_{B,C} + e^*_{add}$

    **else:**

        $G^{(t+1)}_{B,C} \leftarrow G^{(t)}_{B,C} - e^*_{drop}$

    **Let** $f^*_{add} = (u^*_{add}, i^*_{add}) \leftarrow \arg\max \nabla_{X_{fake}}[J(A', X') - c * L(D(X'))]$

    $f^*_{drop} = (u^*_{drop}, i^*_{drop}) \leftarrow \arg\min \nabla_{X_{fake}}[J(A', X') - c * L(D(X'))]$

    **if** $|\nabla_{X_{fake}}[J(A', X') - c * L(D(X'))]_{f^*_{add}}| > |\nabla_{X_{fake}}[J(A', X') - c * L(D((X'))]_{f^*_{drop}}|$ **:**

        $G^{(t+1)}_{X_{fake}} \leftarrow G^{(t)}_{X_{fake}} + f^*_{add}$

    **else:**

        $G^{(t+1)}_{X_{fake}} \leftarrow G^{(t)}_{X_{fake}} - f^*_{drop}$

**return:** $G^{(t)}$

---

$B, C, X_{fake}$ and then 10 iterations of $D$ updates. The Greedy-GAN algorithm is given as Algorithm 2. Greedy-GAN supports both adding and dropping links and features. In the algorithm, we add or drop elements according to the absolute gradient of elements, and the one with larger absolute value will be chosen.

## 4.2 TARGETED ATTACK

Next we extend the proposed algorithms to conduct targeted attack on GCNs. Given an adjacency matrix and a feature matrix, the goal is to make nodes to be classified as a desired class by manipulating links and features of fake nodes. We present methods of attacking the whole dataset and attacking only a single node for different situations.

In our method, when attacking the whole dataset, the fake nodes labels are given by a uniform distribution, which is the same as in the non-targeted attack setting. In targeted attack, we define the objection function as:

$$J(A', X') = \sum_{i=1,\ldots,n} \left( [\hat{A}'\sigma(\hat{A}'XW^{(0)})W^{(1)}]_{i,y^*_i} - \max \left( [\hat{A}'\sigma(\hat{A}'XW^{(0)})W^{(1)}]_{i,:} \right) \right), \quad (7)$$

where $y^*_i$ is the target label for adversarial attack of node $i$. In this objective function, if largest logit of node $i$ is the target label $y^*_i$, the objective value is $0$ ; otherwise the value is negative. Similar to the non-targeted attack method, we would like to find $B$, $C$ and $X_{fake}$ using Greedy attack to solve the optimization problem (4) and Greedy-GAN attack to solve the optimization problem (6).

For attacking a single node, we add three fake nodes with target labels, and the objective function is

$$J(A', X') = [\hat{A}'\sigma(\hat{A}'XW^{(0)})W^{(1)}]_{i,y^*_i} - \max \left( [\hat{A}'\sigma(\hat{A}'XW^{(0)})W^{(1)}]_{i,:} \right), \quad (8)$$

where $i$ is the node to attack, then update their edges and features by Greedy attack. We do not perform Greedy-GAN attack due to the number of fake nodes is too small, the sample of real nodes and fake nodes are extremely unbalanced, which leads to over-fitting and an inaccurate discriminator.

## 5 EXPERIMENTS

We use Cora and Citeseer attribute graphs as benchmarks, with a 20% / 80% labeled / unlabeled split of the data. We add some fake nodes with corresponding fake feature matrix, and the rate of

Table 1: Accuracy of GCN before and after non-targeted attacks. Note that the final two rows are the F1 score of the discriminator—lower values indicate that added fake nodes are harder to be detected.

| Dataset | Cora | | Citeseer | |
|---|---|---|---|---|
| Normalization | row-wise | symmetric | row-wise | symmetric |
| Clean | 0.84 | 0.81 | 0.76 | 0.73 |
| Random | 0.29 | 0.29 | 0.33 | 0.31 |
| Greedy | 0.10 | 0.16 | 0.15 | 0.14 |
| Greedy-GAN | 0.12 | 0.34 | 0.25 | 0.36 |
| F1 score for Greedy | 0.65 | 0.75 | 0.83 | 0.72 |
| F1 score for Greedy-GAN | 0.40 | 0.46 | 0.53 | 0.24 |

Table 2: Success rate for targeted attacks on the whole graph.

| Dataset | Cora | | Citeseer | |
|---|---|---|---|---|
| Normalization | row-wise | symmetric | row-wise | symmetric |
| Greedy | 0.99 | 0.97 | 0.90 | 0.85 |
| Greedy-GAN | 0.64 | 0.44 | 0.45 | 0.22 |

fake labels, to investigate the influences on the results. The default number of fake nodes we add is 20% of real nodes, and 25% percent of fake nodes (i.e. 5% of number of real nodes) have labels. In the initial state, the fake nodes have random features only, and no connection to any other node. The results are shown in GCN classification accuracy for non-targeted attacks, and success rate for targeted attacks. Lower classification accuracy or higher success rate indicates more effective attacks.

## 5.1 NON-TARGETED ATTACK

We compare the effectiveness of random, Greedy algorithm, and Greedy-GAN algorithm in attacking a given GCN. Table 1 shows the accuracy before and after non-targeted attacks. We can see that both Greedy and Greedy-GAN methods work with row-wise and symmetric normalization on the adjacency matrix. Greedy reduces the classification accuracy further than Greedy-GAN, because Greedy-GAN generates features of nodes closer to the real nodes, and leads to lost in attacking performance. We train a discriminator using randomly generated nodes, then feed it with fake features generated by Greedy and Greedy-GAN algorithms. The f1 scores of the discriminators shows that it is much harder to differentiate the features of a node between real and fake under attacks by Greedy-GAN, as compared to Greedy.

## 5.2 TARGETED ATTACK

A targeted attack tries to force the GCN to classify nodes to a certain class. We perform two different experiments for targeted attacks: (1) attack the whole dataset; (2) attack only a single node.

When attacking the whole dataset, we randomly select a class, and other settings are the same as non-targeted attacks. Table 2 shows that targeted attacks work for both Greedy and Greedy-GAN algorithms. In particular for Greedy attack, we can achieve effective attacks, while Greedy-GAN will make fake nodes indistinguishable from real ones.

When attacking only a single node, we add three fake nodes with target labels. Table 3 shows the average success rate of attacking one node is very high. Due to the tiny number of nodes added, the change in distributions of labels is unnoticeable. In our experiments, we add 13 edges and 10

Table 3: Success rate for targeted attacks on a single node.

| Dataset | Cora | | Citeseer | |
|---|---|---|---|---|
| Normalization | row-wise | symmetric | row-wise | symmetric |
| Greedy | 0.94 | 0.83 | 0.80 | 0.62 |

Table 4: Accuracy under Greedy attack, for nodes with different degrees; Cora Dataset with symmetric normalization.

| nodes degree | (0,5] | (5,10] | (10,20] | (20, $\infty$) |
|---|---|---|---|---|
| clean | 0.82 | 0.85 | 0.87 | 0.75 |
| Greedy | 0.13 | 0.25 | 0.37 | 0.5 |

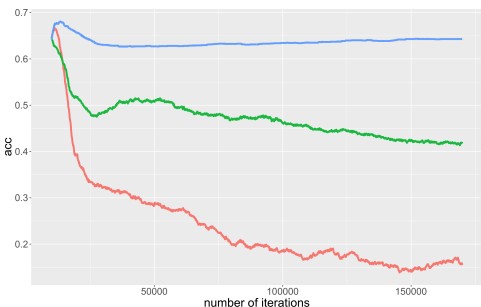 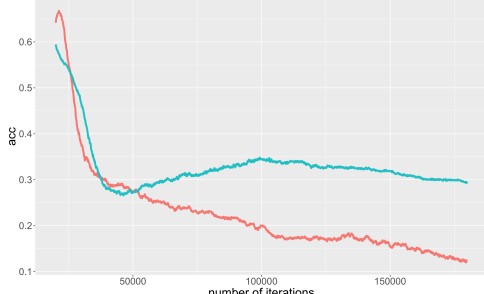

Figure 1: Accuracy of GCN by attacking features, edges or both, with Cora dataset. Green: attacking edges; blue: attacking features; red: attacking both.

Figure 2: Accuracy of GCN using row-wise or symmetrical normalization, with Cora dataset. Red: Row-wise normalization, Green : Symmetric normalization

features per fake nodes on average. We did not perform Greedy-GAN attack, because the numbers of real and fake nodes are so unbalanced that over fitting will make the discriminator inaccurate.

We notice that attacking nodes in certain classes are more difficult than others. For example, attacking nodes in class 0 (249 nodes) from Citeseer dataset (3312 nodes in 6 classes) is hard. The percentage of class 0 nodes are lower that others, thus the attacked node may only have weak linkage to class 0, and the classification is more influenced by nodes in other classes. Only linking the attacked node with three fake nodes is not powerful enough to make it classified as class 0.

### 5.3 PARAMETER SENSITIVITY

**Degree of Nodes** As shown in Table 4, nodes with smaller degrees are easier to attack, i.e. produce a different classification from the original one. When adding extra links between fake nodes and real nodes, higher degree nodes are more resistant to the impact of fake nodes.

**Features vs. Edges** We want to know which is more important in influencing the output of the GCN: modifying features or modifying edges. In this experiment, we randomly initialize some edges and features for the fake nodes, then update either features or edges, while keeping the other one fixed. Figure 1 shows that modifying features have a minor effect on the classification accuracy, and the attack becomes less effective under too many feature modifications. Edges play a more significant role for attacking. This insight inspires us to use a GAN to generate fake nodes and features.

**Row-wise vs. Symmetrical Normalization** We found that the implementations of GCNs may differ by frameworks in how they normalize the adjacency matrix. The Pytorch code normalizes the adjacency matrix by row $\hat{A} = D^{-1}(A + I)$, while the TensorFlow code normalizes it symmetrically $\hat{A} = D^{-1/2}(A + I)D^{-1/2}$. They are both correct as claimed by the authors, and this is an ambiguous point in the definition of GCNs in general.

In this experiment, a fixed amount of fake data are added per iteration, i.e. more iterations means more fake data. Figure 2 shows that when training a "clean" GCN, with no attack, there is only a slight difference in performance, and row-wise normalized GCN is better than the symmetric normalized one. While under attacks by a small amount of fake data, row-wise normalized GCN is more robust than the symmetric normalized one. However, when under attacks by a large amount of fake data, the symmetric scaling version is more robust. The reason might be that, when adding a fake link, the change of each element in adjacency by the symmetric normalization is smaller than the row-wise one.

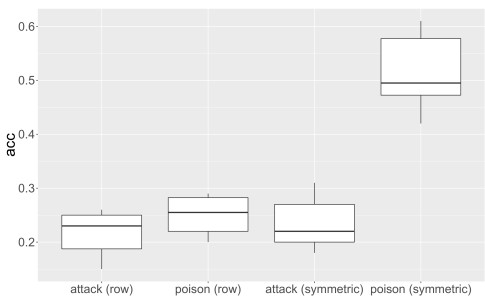
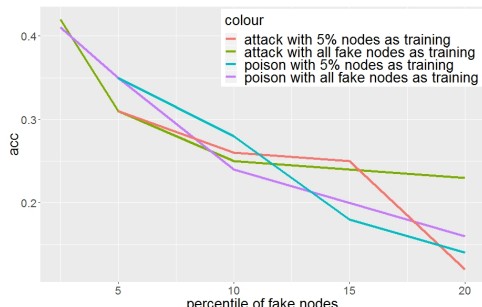

Figure 3: Accuracy of GCN under evasion or poisoning attacks, with Citeseer dataset.

Figure 4: Accuracy of GCN using different percentage of fake nodes, with Cora dataset.

Table 5: Accuracy of GCN under non-attacks by Greedy and Greedy-GAN, with different label rates and Cora dataset.

| label rate | 0% | 5 % | 10 % | 15 % | 20 % |
|---|---|---|---|---|---|
| Greedy | 0.08 | 0.10 | 0.18 | 0.21 | 0.23 |
| Greedy-GAN | 0.33 | 0.12 | 0.13 | 0.14 | 0.16 |

**Evasion vs Poisoning** In industry, GCN is normally used with longitudinal data. Training a new network is time consuming, and changes between updates are not significant, thus people just retrain the network after certain amount of time. Figure 3 shows the difference in classification accuracy without (evasion) and with (poisoning) retraining including the fake data in the process. The GCN is retained with learning rate 0.01, 50 epochs after the data has been modified. We run GCN with different random seeds. The results show that both evasion and poisoning can effectively reduce the accuracy of GCN, and the symmetric normalization is more robust under poisoning attacks.

**Number of Fake Nodes** Figure 4 shows how the number of fake nodes influences the classification accuracy. We use 2.5%, 5%, 10 % and 20% of fake nodes, and assign random labels to the fake nodes. As expected, more fake nodes yields more effective attacks, for both evasion and poisoning. We notice with 20 % of fake nodes and all nodes labeled, the attack has lower effectiveness than the default setting with 20 % fake nodes and 5 % labeled. We will talk about this in the next section.

**Label Rate** We will discuss the effect of different label rates of fake nodes. In Table 5 we keep the percentage of fake nodes to 20%, and change the percentage of labeled nodes from 0 % to 20%. Intuitively we thought that a higher label rate will lead to a larger impact in GCN's accuracy. However, it turns out that when the label rate is 0%, non-targeted attack (Greedy) has the largest effect. We suspect the reason may be that when the label rate is 0, i.e. not considering the features matrix, multiple edges might be viewed as adding edges for the existing nodes in the graph. It is also true for targeted attack; for example, if we give all the fake nodes random labels for the Citeseer dataset and use a symmetric normalized adjacency matrix, the targeted attack success rate is 0.81, compared to 0.85 when with label as shown in Table 2. For attack by Greedy-GAN, a different pattern is observed. Using 0 label rate yields the least effect, and from 5% to 20%, the result is similar to what we find using Greedy.

## 6 CONCLUSION

We present two algorithms, Greedy and Greedy-GAN, on adversarial attacks of GCNs by adding fake nodes, without changing any existing edges or features, for both non-targeted and targeted attacks. We successfully attacked existing GCN implementations, and explored parameter sensitives, such as number of fake nodes and different label rates of fake nodes. To make the attack unnoticeable, we added a discriminator using the Greedy-GAN algorithm to generate features of fake nodes. We noticed that data cleaning before training is crucial, and adding a discriminator makes the impact of attacks weaker. There is a trade-off between the efficiency of attack and realness of fake nodes' features.

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
