# OpenReview forum: "ATTACK GRAPH CONVOLUTIONAL NETWORKS BY ADDING FAKE NODES"
_ICLR.cc/2019/Conference_

### Official Review · AnonReviewer2 · 2018-11-01
**Nice but straightforward idea to attack graph CNNs; paper not always well-written**

**Rating:** 3
**Confidence:** 2

**Review:**

The main idea of this paper is that a 'realistic' way to attack GCNs is by adding fake nodes. The authors go on to show that this is not just a realistic way of doing it but it can done in a straightforward way (both attacks to minimize classification accuracy and GAN-like attacks to make fake nodes look just like real ones).

The idea is neat and the experiments suggests that it works, but what comes later in the paper is mostly rather straightforward so I doubt whether it is sufficient for ICLR. I write "mostly" because one crucial part is not straightforward but is on the contrary, incomprehensible to me.  In Eq (3) (and all later equations) , shouldn't X' rather than X be inside the formula on the right? Otherwise it seems that the right hand side doesn't even depend on X' (or X_{fake} ).
But if I plug in X', then the dimensions for weight matrices  W^0 and W^1 (which actually are never properly introduced in the paper!) don't match any more. So what happens? To calculate J you really need some extra components in W0 and W1. Admittedly I am not an expert here, but I figure that with a bit more explanation I should have been able to understand this. Now it remains quite unclear...and I can't accept the paper like this.

Relatedly, it is then also unclear what exactly happens in the experiments: do you *retrain* the network/weights or do you re-use the weights you already had learned for the 'clean' graph?

All in all:
PRO:
- basic idea is neat
CON:
- development is partially straightforward, partially incomprehensible.

(I might increase my score if you can explain how eq (3) and later really work, but the point that things remain rather straightforward remains).

---

### Official Review · AnonReviewer3 · 2018-11-02
**An interesting idea, but the improvement over existing work is unclear**

**Rating:** 3
**Confidence:** 4

**Review:**

The authors propose a new adversarial technique to add “fake” nodes to fool a GCN-based classifier. The basic approach relies on a greedy heuristic to add edge/node features, and the authors also present a GAN-based approach, which allows the model to add “fake” nodes that are not easily distinguishable from regular nodes. The primary motivation behind the idea of adding “fake” is that it is unrealistic to change the features/edges of existing nodes. Experimental results show that adding a large number (20% in most cases) of fake nodes can significantly degrade accuracy of a GCN, and results show that the GAN-based approach is somewhat effective at making the “fake” nodes less distinguishable.  In terms of strengths, the GAN-based approach is well-motivated and it appears that the authors were thorough in their experiments on Cora/Citseer (e.g., with a number of ablation/sensitivity studies).

However, while interesting, this paper has a number of areas where it could be substantially improved:

1) With regards to the motivation: It is not clear what substantive technical novelty there is in the idea of “adding fake nodes”, compared to existing approaches that simply modify existing nodes in an adversarial way. Intuitively, the approach of Zugner et al can already handle this case of "adding new nodes". One just adds a set of nodes with random/null edges/features to the graph, treats this as their “attacker node” set and then runs Zugner et al's greedy algorithm. Some clarification on why this simple application of Zugner et al's approach does not work would be useful and/or empirical results using their method as a baseline would be useful. (Also, Zugner et al was published in KDD 2018, so the citation should be corrected).

2) In Zugner et al, they derive approximations and algorithms that allow them to compute the score of adding/removing an edge in constant time. The greedy approach in this work appears quite expensive as every greedy update requires an expensive gradient computation. Some discussion of computational complexity would improve the paper.

3) Results are only provided on two small datasets (presumably due to the large computational cost for the approach). These two very small datasets are not indicative of many real-world scenarios, and additional results on larger (and more diverse) datasets would greatly strengthen the paper.

4) Adding 20% fake nodes seems like a prohibitively large number. Even 5% fake nodes is extremely large. It is unclear what real-world applications could admit such drastic numbers of fake nodes, and some comments on this would greatly strengthen the paper.

5) The GAN method is interesting and well-motivated, but it is not clear if this method offers any utility beyond the “distribution matching” approach of Zugner et al (Section 4.1 of their paper). A comparison between these methods is necessary to justify the utility of the proposed GAN-greedy approach.

---

### Official Review · AnonReviewer1 · 2018-11-07
**Should be supported with stronger experiments and be more clearly presented**

**Rating:** 4
**Confidence:** 3

**Review:**

This paper presents an idea of adding fake nodes to attack a graph network model, by a GAN style trainning procedure.

However I concern about the experimental parts, which are only evaluated on small settings.

Plus, the notations are inconsistant, whereas the objective function in (3) has nothing to do with $X_{fake}$. I tend to believe that this should be a typo.

The greedy optimization should generally be highly costed, although it works well for learning sparse representation in previous literature, however, in the graph setting, I am not sure that this is a good fit for $O(|V|^2)$ variables. Perhaps the author need to argue why this is efficient, or to propose other methods.

---

### Meta-Review · Area_Chair1 · 2018-12-17
**reject**

**Confidence:** 4
**Recommendation:** Reject

**Metareview:**

While the main idea of the paper is nice, the reviewers are not satisfied with the clarity of the material and the execution.